# A Novel Approach for Measuring the Thickness of Refractory of Metallurgical Vessels

**DOI:** 10.3390/ma13245645

**Published:** 2020-12-10

**Authors:** Yao Ge, Ying Li, Han Wei, Hao Nie, Weitian Ding, Yi Cao, Yaowei Yu

**Affiliations:** State Key Laboratory of Advanced Special Steel, Shanghai Key Laboratory of Advanced Ferrometallurgy and the Science and Technology Commission of Shanghai Municipality, School of Materials Science and Engineering, Shanghai University, Shanghai 200444, China; ge_geyao@hotmail.com (Y.G.); yingli9104@163.com (Y.L.); weihan@shu.edu.cn (H.W.); niehao@shu.edu.cn (H.N.); dingweitian@shu.edu.cn (W.D.); 13166255603@163.com (Y.C.)

**Keywords:** blast furnace monitoring, refractory thickness, vessel campaign, electromotive force, hearth campaign

## Abstract

The advancement of metallurgical vessels, such as blast furnaces, shaft furnaces, and torpedo ladles, can be better controlled and expanded for a greater lifespan if the thickness of the refractory lining wear is known and predicted. In the past, various methods including radioactive tracers, infrared (IR) thermography, electromagnetic waves, ultrasonic tomography, and temperature field have been tested to determine the thickness of the refractory wall. However, for various reasons, these methods have failed to be effective. This paper presents a novel method—electromotive force (EMF)—for predicting the thickness of refractory lining wear in vessels, including a small-scale vessel in the laboratory, an industrial torpedo ladle, and in the two refining hearths of blast furnaces. The experimental results show that the magnitude of the EMF signal increases with a decrease in wall thickness. Prediction values of the refractory wall thickness are consistent with measured ones. The relative error of EMF measurement for the torpedo ladle is around 6.8%. The EMF measurement of blast furnace hearths is quite accurate, and the relative error is less than 11%.

## 1. Introduction

Refractory materials are designed to remain physically and chemically stable under high-temperature conditions and are used for the linings of furnaces [1], kilns, ladles [2], reactors, converters, and other types of high-temperature vessels [3,4]. For instance, blast furnace hearths are lined with graphite, ramming material, ceramics, and carbon bricks [5] from the core to the periphery. Fracturing, damage, corrosion, and deeper deterioration of the hearth refractory are caused mainly by mechanical and thermal stress cracks, and chemical attacks which result in the loss of heat transfer capability, shrinkage, and a change in thermal conductivity of the refractories [6,7,8]. Furthermore, a sudden shutdown of the blast furnace [9] causes a loss in the production of hot metal and shortens the lifespan of the refractory materials. Therefore, the hearth advancement may be better controlled if the thickness of the refractory material can be measured accurately.

In the past, varying approaches have been proposed to detect and measure the thickness of hearth linings [10]. Short half-life radioactive tracers dumped at the top of the furnace have been used to detect the thickness of linings [11,12,13]. The failure in the tracer method is that the hearth linings of blast furnaces are nonuniform and refractories with different densities and material properties are utilized in the lining. IR thermography has been considered as a method to measure hearth lining wear. However, steel shell oxidization affects the surface emissivity which results in inaccuracies in the IR thermograph measurements [13]. By detecting electromagnetic waves, such as microwaves [14] and radar, refractory thickness can be determined. However, electromagnetic waves cannot pass through the steel shell and enter the lining. In the late 1990s, Ultrasonic tomography (UT) and acoustic ultrasonic-echo (AU-E) techniques developed by the Hatch NDT Group were also tested to measure refractory thickness wear [15,16]. However, the stress pulses in AU-E and waves in UT increased the measurement error twice by the “go” and “return” routes of signal transmission. Analysis of temperature distribution by thermocouples and heat fluxes in refractory lining is a rudimentary technique that has been used to determine the refractory wear profile in furnaces [17], and was validated by dissection data at the end of more than a 10-year period [18,19]. In recent years, scientists proposed a variety of erosion mathematical models and expert systems [20,21] to estimate the profile of the erosion by the thermocouple’s data. These mathematical models are based on heat conduction in solid material by measuring the temperatures of the thermocouples in the refractories [22]. Therefore, the coefficient of heat conductivity of refractories is dependent on the temperature and directly affects the model results.

In the present work, a novel method of electromotive force (EMF) technology based on the thermoelectric effect is proposed for the first time to measure the thicknesses of refractory wear at the laboratory and plant scales.

## 2. Experimental Methods

In order to apply the EMF technology to the measurement of thicknesses of refractory wear, three experiments were designed. Laboratory scale experiments, first, were completed on two graphite crucibles where EMF sensors were installed in the different locations of the vessel. Second, a torpedo ladle experiment was performed on the torpedo ladle of a steel plant where EMF sensors were installed in the air exhausting holes of the shell. Third, the blast furnace hearth experiments were carried out using a new blast furnace.

### 2.1. Experimental Apparatus and Procedure of Laboratory Scale Experiment

The laboratory scale experimental apparatus was composed of refractory mortar and two graphite crucibles as shown in Figure 1, where the iron lines are EMF sensors and thick bolts are inserted in the wall of the crucible to stabilize the thin molybdenum alloy lines around the circumference of the vessel.

The three scenarios of the laboratory vessel used to simulate the erosions of hearth wear are presented in Figure 2. Figure 2a illustrates three arcs with different heights (5, 10, and 15 mm) made by the aluminum oxide. The arcs were inserted and fixed along the inner wall at the bottom of the vessel before the experiments were performed. Another new vessel had four thicknesses (5, 10, 15, and 20 mm) at different points along the circumference, which can be seen in Figure 2b. EMF sensors were fixed at five depths (24, 19, 14, 9, and 4 mm) in a hole on the wall of the vessel as shown in Figure 2c, and they were used to measure the effect of EMF signals on the thickness of the wall.

At the beginning of the experiments, around 5 kg of liquid tin was dumped into the apparatus from the entrance as shown in Figure 1. Then, the tin flowed out with the gravitational force from the exit at the bottom of the vessel. During the drainage of the liquid metal, the measurement system shown in Figure 3 includes the EMF sensors, the signal transform, the signal visualization, and a laptop that recorded the EMF signals.

A relative value of the EMF (ΔEMF [9,23]) was calculated by the difference between the signals at various heights and the basic value at the bottom of the vessel:(1)$$\Delta \mathrm{EMF}={\mathrm{EMF}}_{\mathrm{base}}-{\mathrm{EMF}}_{\mathrm{top},\mathrm{middle}\;\mathrm{or}\;\mathrm{bottom}}$$

### 2.2. Experimental Apparatus and Procedure of Industrial Torpedo Ladle Experiments

In order to validate the new approach, a 285 t torpedo ladle was used in the industrial experiment to measure the thickness of the refractory lining wear by EMF signals for 3 days. The refractory properties of the torpedo ladle wall, the locations, and the depth of EMF sensors are shown in Figure 4A, where the “a” has two sensors at 32 and 44 mm depths, respectively, and the depth of the other sensors is 32 mm. All sensors were fixed in pyrophyllite brick by bolts. The designed thickness of each layer in the torpedo ladle is shown in Figure 4B.

Before the torpedo ladle experiment, the torpedo ladle first arrived at the hot-metal receiving position under the cast house. Then, during the process of receiving the hot metal, the torpedo ladle measurement system, like the one in Figure 3, was installed to collect EMF signals.

Relative EMF signals from the different measurement locations on the torpedo ladle were calculated with Equation (2):(2)$$\Delta \mathrm{EMF}={\mathrm{EMF}}_{\mathrm{base}}-{\mathrm{EMF}}_{\mathrm{deep}34\;\mathrm{or}\;44}$$

### 2.3. Experimental Apparatus and Procedure of Industrial Blast Furnace Hearth

The EMF method was also applied to the new hearth of a blast furnace with 2000 m^3^ volume, where an industrial experiment was performed to measure the thickness of refractory wall.

An EMF measurement system was designed according to the structural characteristics of the blast furnace’s hearth and the experiments in the previous laboratory and torpedo ladle. It included an EMF sensor, compensation wire, junction box (installed on the blast furnace shell), wiring bridge, signal converter, and a display device as shown in Figure 5. The sensors contained molybdenum wire (the core), silicon dioxide material for insulation, and inconel material for protection (the surface). The EMF sensors, made of molybdenum alloy wire, were buried into the graphite brick during construction of the hearth of the furnace as shown in Figure 6. They were inserted at three different depths in the refractory via a hole along two directions of the hearth at 8494, 9494, and 10,994 mm, respectively (see Figure 7). In order to form a close circuit [23,24,25] in the hearth, a reference point was chosen at the level of 6394 as shown in Figure 7. The wires of the EMF sensors sent the signal to the junction box. The compensation wire connected the junction box and the signal converter through the wiring bridge on the blast furnace platform. Finally, the signals of EMF were displayed on the monitor in the main control room of the blast furnace operation.

The above installation of the EMF hardware was completed during the construction of the blast furnace. After the blast furnace was blown in for 3 months to achieve stable operation and the desired design production capacity, the experiments were conducted in 3 days.

## 3. Results and Discussion

### 3.1. Laboratory Scale Experiments

In order to analyze the relationship between EMF signals and refractory thickness, three laboratory experiments were designed. Figure 8a shows the EMF signals of the three heights of the arcs in the bottom of the crucible during the draining process of the molten tin. Because the contact surface between the arc and the inner wall of the vessel was fragile, EMF increased weakly with the increases of the heights of the arcs after 10 s, and there was no good correlation with the segment level of arc, especially during the first 10 s.

Another laboratory experiment was performed on a new vessel, which had four different thicknesses (5, 10, 15, and 20 mm) at different points along the circumference. The results are shown in Figure 8b. With an increase in the wall’s thickness, the EMF decreased, especially after 9 s. This indicated that the EMF signals had a good correlation with the thickness of the wall. Furthermore, the differences in EMF (ΔEMF) were almost equal among the different thicknesses of the wall.

The third way to study the relationship between the thickness of the refractory material and the EMF signal was to use the EMF sensors that were fixed at five depths (24, 19, 14, 9, and 4 mm) in a hole on the wall of the vessel. These results are presented in Figure 8c, where the X-axis (time of molten tin drainage) = 0 and the end of the X-axis indicates the start and end of the drainage in the vessel. The EMF signals increased with an increase in the depth of the hole, and the difference in ΔEMF signals was almost equal, especially in the interval of 0 to 6 s shown in Figure 8c. The above indicates EMF signals had a positive relationship with the thickness of the refractory wall.

From the above analysis, EMF signal magnitude can be used to represent the thickness of crucible walls. Therefore, industrial scale experiments were planned on torpedo ladles and blast furnaces, which are described in the next section.

### 3.2. Industrial Torpedo Ladle Experiments

Torpedo ladle measurement experiments were conducted under the casting house of a blast furnace in Baosteel. Figure 9 shows the ΔEMF signals collected from the “a” location during a time period. The EMF signals were collected with the aforementioned receiving procedure of the liquid metal, which showed an increasing tendency over time. Furthermore, ΔEMF at a depth of 44 mm is greater than at a depth of 32 mm, which is consistent with the results obtained from the laboratory vessel.

According to Fourier’s law of heat conduction [20,26], the refractory thickness of each layer is calculated by EMF, listed in Table 1. The thickness of each layer has the same value as displayed in Figure 4B, except for the Al_2_O_3_-SiC-C (ASC) brick. The ASC brick had a thickness of 130.35 mm from EMF calculation, and it was 140 mm from the engineer’s measurement. Both values were almost half that of the design value (260 mm). Furthermore, the relative error of EMF was only 6.8%.

### 3.3. Industrial Blast Furnace Hearth Experiments

In order to further study EMF monitoring technology, a 3-day monitoring experiment was carried out on a newly constructed blast furnace. EMF signals on the blast furnace included the basic, the periodic, and the noise part [27]. The noise should be removed. When the blast furnace operation was stable, the EMF value fluctuated in a period, as shown in the normal noise in Figure 10a. Moreover, when there were disturbances, the amplitude of the EMF signal decreased, which can be seen by the special noise shown in Figure 10a. The removal method of the noise signals is further explained in our previous work [28].

The thickness of the hearth wall can be seen in Table 2. According to the installation points of the EMF sensors and the structure of the blast furnace, the EMF measurement principle [29,30,31] of the blast furnace is drawn in Figure 11 and the equivalent circuit of EMF signals is illustrated in Figure 12. The EMF signal gradually decreased with increasing tapping time. During the tapping gap, the EMF signal strengthened as the liquid iron level in the hearth increased. The absolute value of EMF at 200 mm depth was greater than at 100 mm depth at 8494 and 9494 levels, indicating that EMF increased with the decrease in the refractory thickness. More details of the studies on the effects of liquid levels with EMF can be found in our previous work [28].

In summary, the change in EMF signal was consistent with the previous laboratory and torpedo ladle experiment. Average values of EMF signals at an 8494 level were 0.0082 mv at a 100 mm depth and 0.019 mv at a 200 mm depth, and 0.047 mv at a 100 mm depth and 0.061 mv at a 200 mm depth at a 9494 level.

The calculation of refractory thickness for the 9494 level of blast furnace is described in the following texts:

(1)With regard to the purple loop, Ohm’s and resistance’s law are displayed in Figure 11. Equations (3) and (4) can calculate EMF of the slag-iron liquid at level 8494.
(3)$$\left(\frac{U_{200}}{R6}+\frac{U_{100}}{R7}\right)\times \left(R2+R3+R4+R8+R16\right)+U_{200}=U_{8494}$$
(4)$$R_{i}=\frac{\rho_{i}L_{i}}{S}$$
where $R_{i}$ defines the resistance of the ith material in Figure 11, and $L_{i}$ and *S* express the thickness of the ith material in Figure 11 and the cross-sectional area of the current flow, respectively.(2)The tapping gap between two serial tapping times was very short. Therefore, the liquid level of the hearth could not drop below 9494 mm, under which molten iron and slag exist. As the distance between the EMF sensors at 8494 mm and the one at the bottom of the hearth was 500 mm, and the distance between the 9494 level and the one at the bottom was 1500 mm, the relationship of Equation (5) was satisfied. A relationship of slag-iron liquid between the 8494 level and 9494 level could then be made:
(5)$$U_{9494}=3U_{8494}$$(3)According to the blue loop in Figure 11, the remaining thickness of the refractory material at the 9494 level was predicted by
(6)$$\left(\frac{U_{200}^{\prime }}{R6+R14}+\frac{U_{100}^{\prime }}{R7+R15}\right)\times \left(R10+R11+R12+R8+R16\right)+U_{200}^{\prime }=U_{9494}$$

According to Equations (3)–(6), the thickness of refractory walls at the 9494 mm level was calculated:(7)b=R12ρ12=502.3 mm
where ρ12 is the same as ρ12 and the design thickness is 560 mm.

The relative error of the thickness between the EMF calculation (502.3) and design value (560) was 10.3%. This error was in the allowable range, thus it can be accepted that the relative error of the refractory thickness in the “1” direction is around 12.1%. Therefore, EMF measurement is a suitable method for detecting the thickness of blast furnace hearth walls.

## 4. Conclusions

The use of metallurgical vessels, such as blast furnaces, shaft furnaces, and torpedo ladles, can be better controlled and expanded for a longer lifespan, if the thickness of the refractory lining wear is known and predicted reliably. This work presents a novel method to study refractory lining wear in metallurgical vessels by focusing on the measurement of the EMF signal relationship with the thickness of refractory walls on a small-scale vessel, an industrial torpedo ladle, and a new blast furnace hearth. The main objectives of this study were to observe the correlation between wall thickness and EMF signals from four different types of wall structures. The main conclusions can be drawn as follows.

An EMF approach was built and used to measure the thickness of refractory walls in laboratory scale vessels, industrial scale torpedo ladles, and a new blast furnace hearth;The thickness of refractory walls increased with the decrease in EMF signal magnitude;The comparison of wall thickness obtained by EMF and engineering measurement from a plant showed that EMF could measure the thickness of torpedo ladles with a relative accuracy of 6.8%;For the industrial blast furnace application, it was proven that the EMF is accurate for measuring hearth wall thickness, and the relative error is less than 11%.

## Figures and Tables

**Figure 1 materials-13-05645-f001:**
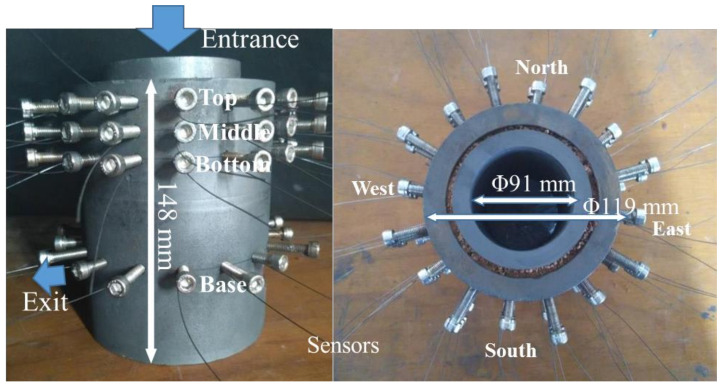
Schematic diagram of a laboratory vessel mimicking blast furnace hearth (left: front view, right: top view, and thin molybdenum alloy lines: electromotive force (EMF) sensors).

**Figure 2 materials-13-05645-f002:**
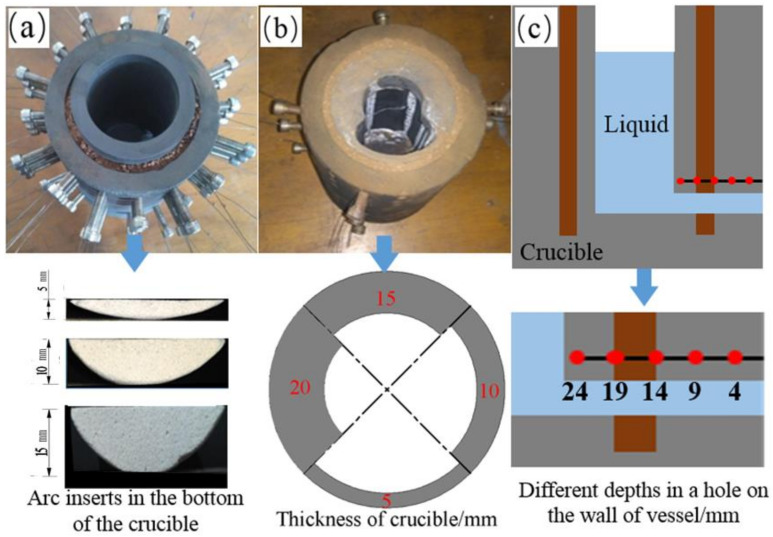
Three scenarios for crucible erosions of the laboratory vessel: (**a**) three arc inserts, (**b**) four different thicknesses along the circumference, and (**c**) different depths (24, 19, 14, 9, and 4 mm) in a hole on the wall of the vessel).

**Figure 3 materials-13-05645-f003:**
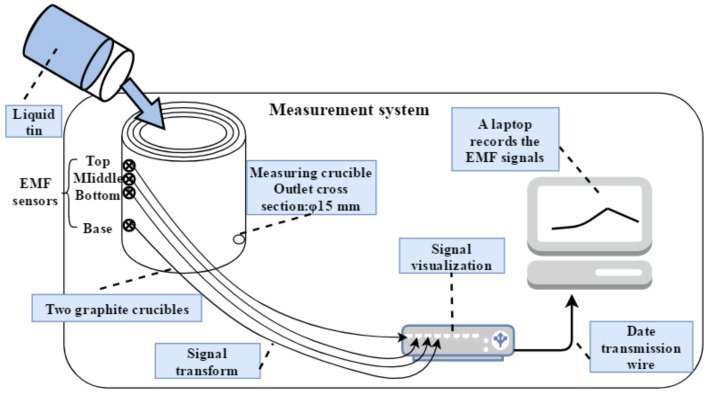
The EMF measurement system from the laboratory.

**Figure 4 materials-13-05645-f004:**
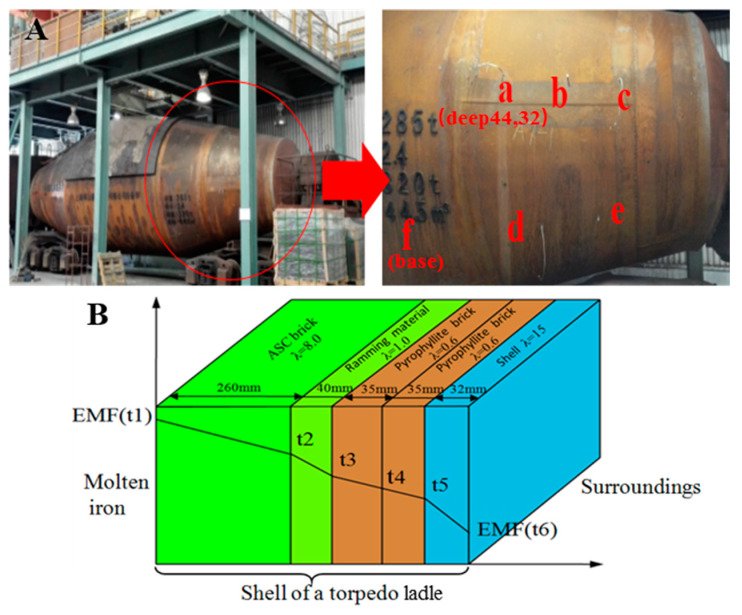
EMF experiments and sensor locations in the shell of a 285 t torpedo ladle: (**A**) EMF locations of *a*, *b*, *c*, *d*, *e*, and *f* on the shell and (**B**) properties of each refractory layer from Al_2_O_3_-SiC-C (ASC) brick to steel shell).

**Figure 5 materials-13-05645-f005:**
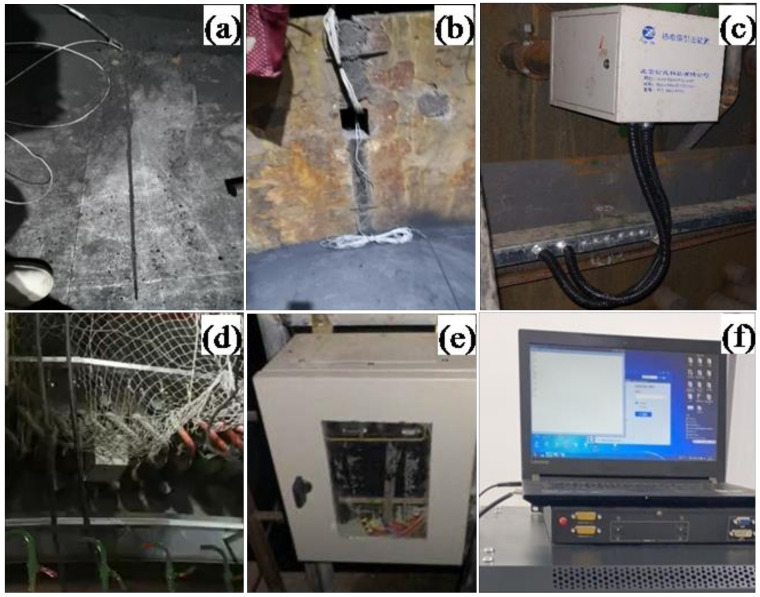
EMF measurement system on the blast furnace hearth: (**a**) EMF sensor, (**b**) compensation wire, (**c**) junction box, (**d**) wiring bridge (**e**) signal conversion box, and (**f**) display device.

**Figure 6 materials-13-05645-f006:**
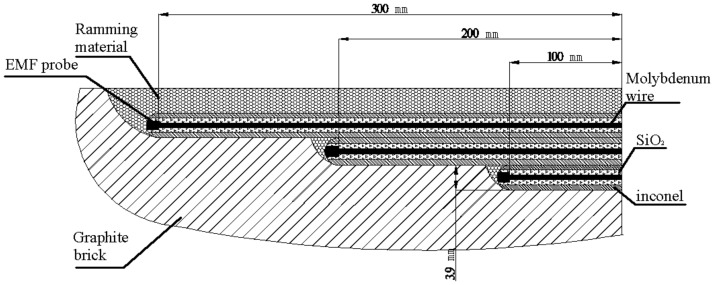
Three EMF sensors in a hole of graphite brick.

**Figure 7 materials-13-05645-f007:**
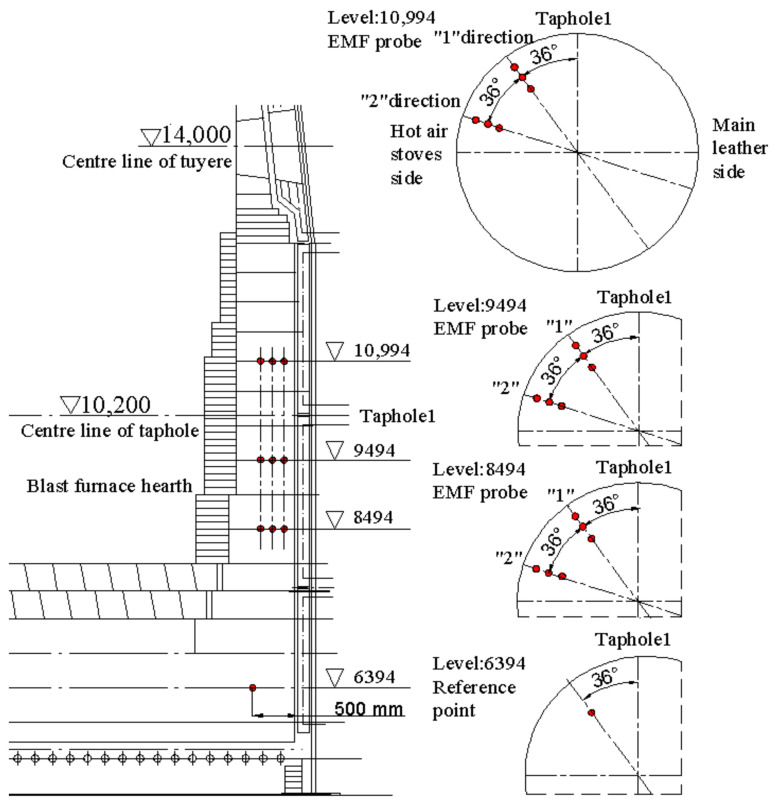
EMF sensors (signals) on the refractory wall of a new blast furnace hearth (left: front view of EMF locations at level 6394, 8494, 9494, and 10,994, right: top view of EMF locations in each levels).

**Figure 8 materials-13-05645-f008:**
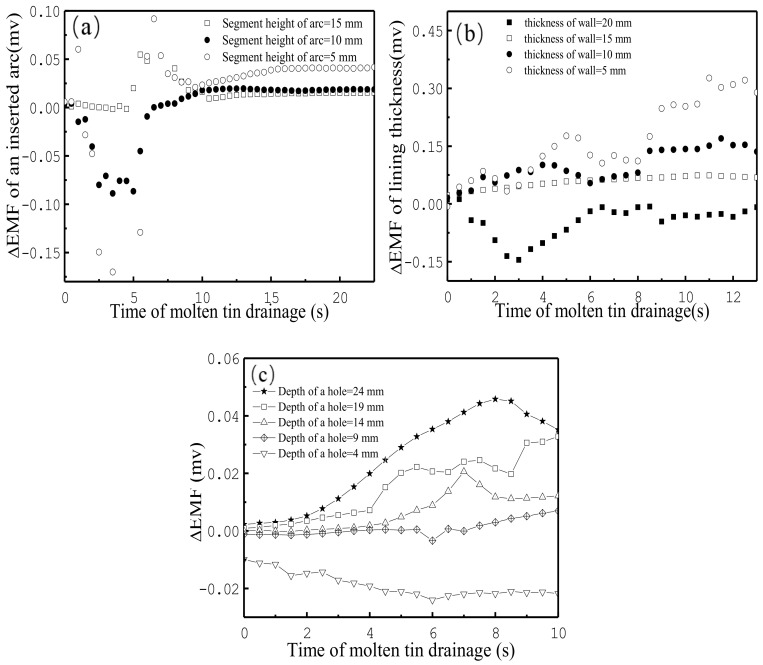
Results of three different scenarios: (**a**) three arc inserts; (**b**) four different thicknesses along the circumference; and (**c**) five depths in a hole in the wall of the vessel: 24, 19, 14, 9, and 4 mm.

**Figure 9 materials-13-05645-f009:**
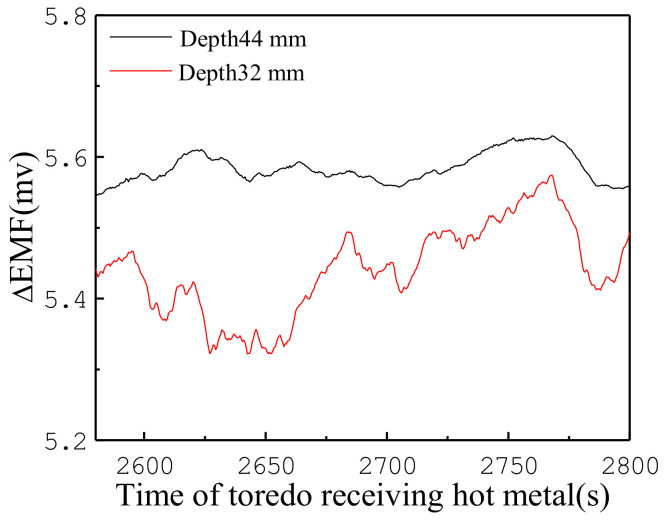
EMF signals on the torpedo ladle refractory wall in a short time interval.

**Figure 10 materials-13-05645-f010:**
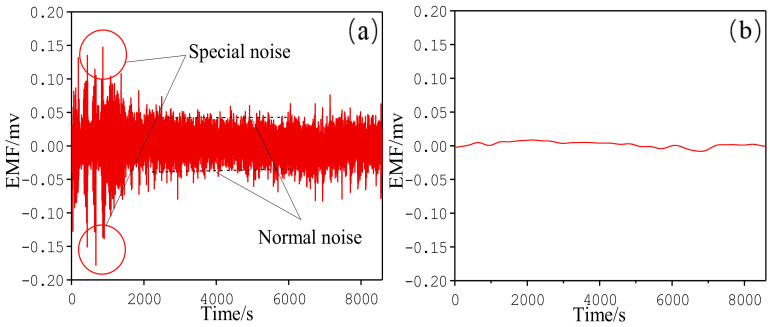
EMF signal: (**a**) original signal and (**b**) processed signal.

**Figure 11 materials-13-05645-f011:**
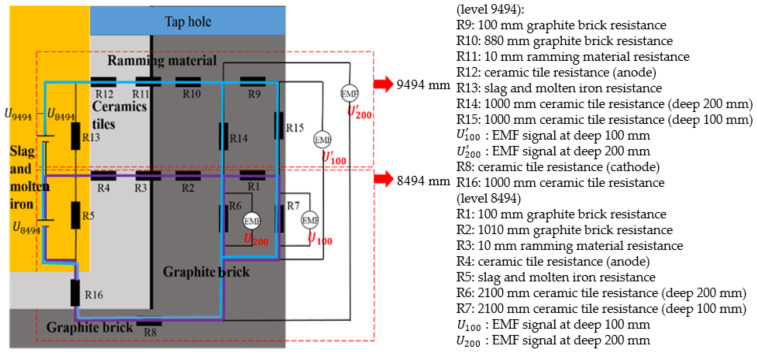
EMF measurement principles of the blast furnace.

**Figure 12 materials-13-05645-f012:**
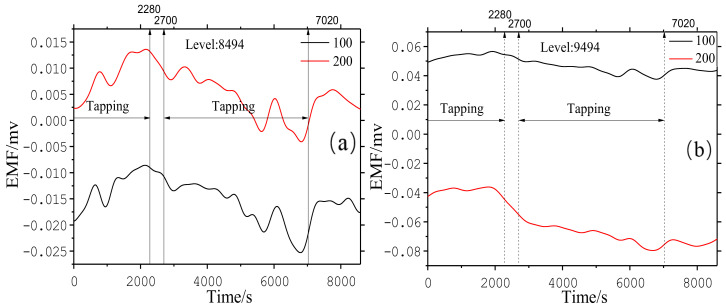
EMF signals of blast furnace hearths: (**a**) EMF signals on the “2” direction at the 8494 level and (**b**) EMF signal on the “2” direction (C.F. Figure 7) at the 9494 level.

**Table 1 materials-13-05645-t001:** Results of torpedo ladle thickness (t1 to t6 reference to Figure 4B).

Locations	Hot Metal (t1)	ASC Brick (t2)	Ramming Material (t3)	Pyrophyllite Brick (t4)	Shell (t5)	Shell (t6)
Thickness (mm)	-	130.35 (140 **)	40.00	35.00	35.00	32.00
Temperature at t1–t6 (°C)	1150 *	1054.39	819.69	477.41	135.14	131.22
EMF at t1-t6	46.99	43.5	34.09	19.5	5.5	5.34
heat flux (J/s)	-	5867.60	5867.60	5867.60	5867.60	5867.60

where * expresses solidification temperature of hot metal = 1150 °C and ** defines the measurement thickness by plant engineers.

**Table 2 materials-13-05645-t002:** Refractory material properties of the hearth as shown in Figure 7.

Materials	Ceramics Tiles	Ramming Material	Graphite Brick
Thickness at level 8494 (mm)	860	10	560
Resistivity (Ω·mm)	2.63 × 10^−8^	0.8	0.577
Thickness at level 9494 (mm)	730	10	560
Resistivity (Ω·mm)	2.63 × 10^−8^	0.8	0.577

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
