# Peer review of "A Novel Approach for Measuring the Thickness of Refractory of Metallurgical Vessels"

_materials, 2020, doi:10.3390/ma13245645_

Round 1
Reviewer 1 Report
The Authors presented a new method of assessing the thickness of the ceramic lining in metallurgical devices using the measurement of the electromagnetic waves. They investigated the dependence of electromagnetic signals on the thickness of walls lined with ceramic materials for four different types of wall structures. They indicated that as the size of the EMF signal decreased, the thickness of the refractory lining increased.
References not in accordance with the materials-template.dot in the text and in the list.
A lots of editorial mistakes.
The presented Article complies with the journal's profile Processes or Energies.
Author Response
Reviewer #1
1- References not in accordance with the materials-template.dot in the text and in the list.
Response: Thanks for your suggestion. We have revised the references in accordance with the format required by the journal. But there are also some documents not find doi.
Lines: 245-322
2- A lots of editorial mistakes.
Response: We sincerely thank the reviewer for careful reading. We feel so sorry for our carelessness. We have modified all the editorial mistakes and highlighted in red.
Reviewer 2 Report
The manuscript by Y. Ge et al. attempts to present an innovative EMF strategy for measuring the thickness of industrial vessels. The manuscript is comprehensive and will be interesting for the researchers in the field. The figures have sufficient quality. I’d like to recommend this manuscript for publication in Materials. Nonetheless, there are some critical issues to be solved before the final acceptance and publication:
1- Title is incomplete and must be corrected. “…of metallurgical vessels” should be added to the end of current title.
2- It’s highly recommended to divide “2. Experimental methods” and “3. Results and discussion” sections. Please see Page 2, line 54. All the important experimental details should be carefully mentioned in detail in Section 2 to be easily reproducible for future readers and users. The authors are introducing a new method; thus it’s highly expected that they elaborate the details of measurement process in a separate sub-title.
3- References need careful reconsideration consistent with the journal's guidelines, such as refs. 4, 5, 7, 8, 10, 14, 20, 22, 25.
4- There are several typo and grammatical errors, and should be corrected before the publication. For example:
Line 83: …, especially after 9 s.
Lines 69 and 70: b: four different thicknesses
Line 74: ….. 5 mm, 10 mm and 15 mm, respectively.
Line 78: …. after 10 s.
Line 79: … first 10 s.
Please correct whole the text accordingly. The units must not have connected to the numbers.
Line 99: b: four different thicknesses
Line 106: at 34 and 44 mm depths
Line 195: According to Equations 2-5, …..
Line 198: …. it can be accepted as …..
5- Where did you explain Figure 3a? It seems it’s in between the lines 72 to 79. Please clearly mention it in the text.
6- Lines 86-93: What are the differences between Figures 3c and 3d? Isn’t it better to merge these two figures in a single figure?
7- Line 111, Figure caption 4: Please define A and B instead of top and bottom images. Then, please correct the text in lines 105 and 108 as follows: “Figure 4A” and “Figure 4B” to omit any probable confusions with the symbols a,b,c,d,e,f in Figure 4A, right-side.
8- Line 121: Please add a proper reference here: “According to Fourier's law of heat conduction [???], …..”
9- Line 112 or 123: Please define ASC abbreviation when it’s first appeared.
10- Line 185: Please cite properly the Ohm’s law and Equations 2,3.
Author Response
Reviewer #2
1- Title is incomplete and must be corrected. “…of metallurgical vessels” should be added to the end of current title.
Response: Thanks, we apologize for my carelessness. We have completed the title.
Line: 3
2- It’s highly recommended to divide “2. Experimental methods” and “3. Results and discussion” sections. Please see Page 2, line 54. All the important experimental details should be carefully mentioned in detail in Section 2 to be easily reproducible for future readers and users. The authors are introducing a new method; thus it’s highly expected that they elaborate the details of measurement process in a separate sub-title.
Response: Thanks for your suggestion. The section 2 has been divied into two parts. We have added important experiment details in Section 2 and highlighted in red.
Line: 59
3- References need careful reconsideration consistent with the journal's guidelines, such as refs. 4, 5, 7, 8, 10, 14, 20, 22, 25.
Response: Thanks for your suggestion. We have revised the references in accordance with the format required by the journal. But there are also some documents not find doi.
Line: 253-319
4- There are several typo and grammatical errors, and should be corrected before the publication. For example:
Line 83: …, especially after 9 s.
Lines 69 and 70: b: four different thicknesses
Line 74: ….. 5 mm, 10 mm and 15 mm, respectively.
Line 78: …. after 10 s.
Line 79: … first 10 s.
Please correct whole the text accordingly. The units must not have connected to the numbers.
Line 99: b: four different thicknesses
Line 106: at 34 and 44 mm depths
Line 195: According to Equations 2-5, …..
Line 198: …. it can be accepted as …..
Response: We sincerely thank the reviewer for careful reading. We feel so sorry for our carelessness. We have modified all the editorial mistakes and highlighted in red.
Lines: 146, 81 and 82, 74, 142, 143, 161, 97, 219, 222
5- Where did you explain Figure 3a? It seems it’s in between the lines 72 to 79. Please clearly mention it in the text.
Response: Thanks, we feel sorry for our poor writings. The description of Figure 3a been highlighted in red in the text.
Line: 139
6- Lines 86-93: What are the differences between Figures 3c and 3d? Isn’t it better to merge these two figures in a single figure?
Response: Thanks for your suggestion. There is no difference between Figures 3c and 3d. We have merged two figures in a single figure.
Line: 160
7- Line 111, Figure caption 4: Please define A and B instead of top and bottom images. Then, please correct the text in lines 105 and 108 as follows: “Figure 4A” and “Figure 4B” to omit any probable confusions with the symbols a,b,c,d,e,f in Figure 4A, right-side.
Response: Thanks for your suggestion. We have modified the picture according to your suggestion and and the new picture has been added in line 103.
Lines: 103-106
8- Line 121: Please add a proper reference here: “According to Fourier's law of heat conduction [???], …..”
Response: Thanks for your suggestion. We have add two references in paper.
- Torrkulla, J.; Saxen, H. Model of the state of the blast furnace hearth. ISIJ int. 2000, 40(5), 438-447, doi:10.2355/isijinternational.40.438.
- Du, G.; Chen, L. Two dimensional model of thermal conduction for the hearth and bottom of the blast furnace. Refractories 1999, 33(4), 216-218, doi:JournalArticle/5af24235c095d718d8f03ed3.
Line: 171
9- Line 112 or 123: Please define ASC abbreviation when it’s first appeared.
Response: Thanks, we have corrected the first occurrence of the abbreviation "ASC" to "Al2O3-SiC-C (ASC)".
Lines: 105 and 106
10- Line 185: Please cite properly the Ohm’s law and Equations 2,3.
Response: Thanks, the original text has been changed to “According to the purple loop in Figure 11, Ohm's and resistance’s law. Equation 3 and 4 can calculate EMF of the slag-iron liquid at the level 8494”. Picture 11 has also been modified in the corresponding position.
Lines: 208, 209, 189, 190
Reviewer 3 Report
The paper presents a novel method based on electromotive force for predicting the thickness of refractory in metallurgical vessels.
The work appears well carried out with different scale tests and it is well described in the paper.
I have only few remarks:
- Introduction: I think that it can be expanded by including also works related to the development of mathematical predictive models for the refractory thickness such as the one by Zagaria et al. (Zagaria, M., Dimastromatteo, V., & Colla, V. (2010). Monitoring erosion and skull profile in blast furnace hearth. Ironmaking & Steelmaking, 37(3), 229-234.)
- 2.1 Laboratory scale experiments:
- line 27 --> please change as follows: "Laboratory scale experimental apparatus is composed..."
- line 62 --> please change as follows: "At the beginning..."
- Figure 3 --> please change as follows: "...14,9 and 4 mm and 24, 19, 14 and 9 mm, respectively)
- 2.2 Industrial torpedo experiments:
- Figure 5 --> please correct as follows: "Time of torpedo receiving hot metal(s)"
- 2.3.1 Experimental equipment and installation:
- Figure 6 --> please use the same character size in the caption
- 2.3.2 Calculation of blast furnace refractory thickness:
- Please explain better the first sentence (the end of the sentence is a little bit confusing).
Author Response
reviewer3
- Introduction: I think that it can be expanded by including also works related to the development of mathematical predictive models for the refractory thickness such as the one by Zagaria et al. (Zagaria, M., Dimastromatteo, V., & Colla, V. (2010). Monitoring erosion and skull profile in blast furnace hearth. Ironmaking & Steelmaking, 37(3), 229-234.)
Response: Thanks for your suggestion. We have added the works about mathematical predictive models for the refractory thickness to the section introduction and highlighted it in red. The specific content is "In recent years, scientists proposed a variety of erosion mathematical models and expert systems [20,21] to estimate the profile of the erosion by the thermocouple’s data. The mathematical models are based on heat conduction in solid material by measurement the temperatures of the thermocouples in the refractories [22]. Therefore, coefficient of heat conductivity of refractories is dependent on the temperature and will directly affect the model results".
Lines: 50-55
- 2.1 Laboratory scale experiments:
- line 27 --> please change as follows: "Laboratory scale experimental apparatus is composed..."
- line 62 --> please change as follows: "At the beginning..."
- Figure 3 --> please change as follows: "...14,9 and 4 mm and 24, 19, 14 and 9 mm, respectively)
Response: Thanks, we feel sorry for our poor writings. The above three errors have been modified in the article.
Lines: 67, 84, 162
- 2.2 Industrial torpedo experiments:
- Figure 5 --> please correct as follows: "Time of torpedo receiving hot metal(s)"
Response: Thanks, we have modified the x-axis of Figue 9 (Original Figure 5).
Line: 170
- 2.3.1 Experimental equipment and installation:
- Figure 6 --> please use the same character size in the caption
Response: Thanks, we apologize for my carelessness. We have changed the caption to a uniform character.
Lines: 129 and 130
- 2.3.2 Calculation of blast furnace refractory thickness:
- Please explain better the first sentence (the end of the sentence is a little bit confusing).\
Response: Thanks, the first sentence has been modified to “EMF signal measured on the blast furnace contains the basic signal, the periodic signal and the noise among which the noise should be removed[27]”. Citation 27 is added here, which has a detailed introduction to the source of the signal.
Lines: 181-182
Round 2
Reviewer 2 Report
The authors properly responded my concerns and extensively modified their manuscript for final publication in Materials. It's now acceptable from my viewpoint.